# Synthesis and Characterization of Amphiphilic Diblock Polyphosphoesters Containing Lactic Acid Units for Potential Drug Delivery Applications

**DOI:** 10.3390/molecules28135243

**Published:** 2023-07-06

**Authors:** Tatsuya Sakuma, Kimiko Makino, Hiroshi Terada, Issei Takeuchi, Violeta Mitova, Kolio Troev

**Affiliations:** 1Faculty of Pharmaceutical Sciences, Tokyo University of Science, 2641, Yamazaki, Noda 278-8510, Chiba, Japan; 2Faculty of Pharmaceutical Science, Josai International University, 1 Gumyo, Togane 283-8555, Chiba, Japan; 3Institute of Polymers, Bulgarian Academy of Sciences, 113 Sofia, Bulgaria

**Keywords:** polyphosphoesters, amphiphilic polymers, micelles, drug delivery

## Abstract

Multistep one-pot polycondensation reactions synthesized amphiphilic diblock polyphosphoesters containing lactic acid units in the polymer backbone. At the first step was synthesized poly[poly(ethylene glycol) H-phosphonate–*b*-poly(ethylene glycol)lactate H-phosphonate] was converted through one pot oxidation into poly[alkylpoly(ethylene glycol) phosphate-*b*-alkylpoly(ethylene glycol)lactate phosphate]s. They were characterized by ^1^H, ^13^C {H},^31^P NMR, and size exclusion chromatography (SEC). The effects of the polymer composition on micelle formation and stability, and micelle size were studied via dynamic light scattering (DLS). The hydrophilic/hydrophobic balance of these polymers can be controlled by changing the chain lengths of hydrophobic alcohols. Drug loading and encapsulation efficiency tests using Sudan III and doxorubicin revealed that hydrophobic substances can be incorporated inside the hydrophobic core of polymer micelles. The micelle size was 72–108 nm when encapsulating Sudan III and 89–116 nm when encapsulating doxorubicin. Loading capacity and encapsulation efficiency depend on the length of alkyl side chains. Changing the alkyl side chain from 8 to 16 carbon atoms increased micelle-encapsulated Sudan III and doxorubicin by 1.6- and 1.1-fold, respectively. The results obtained indicate that these diblock copolymers have the potential as drug carriers.

## 1. Introduction

A well-known problem with biodegradable polymers for medical applications is connected with the safety of their degradation products. In this respect, polyphosphoesters, based on poly(ethylene glycol) (PEG) are promising because they are degraded into small compounds: PEG and phosphoric acid, that are known to be non-toxic and are small enough for natural clearance mechanisms. Poly(alkylene H-phosphonate)s with phosphoester repeating units in the backbone are particularly interesting due to their high reactivity, biocompatibility, biodegradability, low toxicity, and structural resemblance to natural biomacromolecules such as nucleic acids. They are especially attractive advanced reactive and functional polymers due to the following advantages [1]: (i) they are highly reactive; (ii) they are water soluble; (iii) the drug-carrying capacity is determined by the highly reactive P-H group in each repeating units; (iv) the chemical immobilization of drugs proceeds under mild conditions; (v) the presence of highly polar P=O group in the repeating units affords the possibility for physical immobilization of drugs; (vi) possibility of hydrophilic/hydrophobic balance control; (vii) they can be regarded as biodegradable (enzymes the dominant component in the degradation process) and biocompatible synthetic polymers; (viii) they can be designed to have nontoxic building blocks; (ix) they can be administrated over a wider molecular weight range because, after hydrolysis, the low molecular PEG and phosphoric acid will be safely excreted; (x) easy to prepare in industrial scale; (xi) they are low toxic (IC_50_ 1000 mg/kg). They are one of the most interesting class polyphosphoesters because both the polymer backbone and phosphorus substituents can be modified. Poly(alkylene H-phosphonate)s offers a unique opportunity to introduce various modifications at the phosphorus center through different reaction schemes [2,3,4,5,6,7,8,9,10,11,12,13,14]. They are being actively investigated for pharmaceutical and biomedical applications such as carriers of drugs [15,16,17,18,19,20,21] and genes [22,23,24,25,26,27,28]. We recently published the synthesis of thermoresponsive amphiphilic polyphosphoesters based on poly(oxyethylene H phosphonate)s with different lengths of poly(ethylene glycol) segments and aliphatic alcohols with different alkyl chain lengths, using polycondensation reactions. They have the potential as carriers of hydrophobic drugs [29].

Lactic acid (LA) is a naturally occurring and renewable monomer that is used for the synthesis of biodegradable polymers. LA-based polymers are important biomaterials for pharmaceutical, medical, and environmental applications because of their biocompatibility and environmental degradability [30,31,32,33,34]. The common way to prepare LA-based polymers is by ring-opening polymerization (ROP) of lactones and lactides [35,36,37]. However, the catalysts used to initiate ROP are difficult to remove from the resulting polymer [38], and the polymer’s purity is quite important for biomaterial applications [39]. In addition, the preparation of lactide is complicated and expensive [40]. Thus, it is of significant merit to develop a new route for preparing LA-based polymers while retaining their degradability. So far, there are no reports that describe LA-based polymers that contain single LA units. Recently, studies on smart pellets for controlled delivery [41] and potential smart colonic drug delivery system [42] were reported. The polymers synthesized in this study are also expected to contribute to the realization of a more advanced drug delivery system.

Herein we report a multistep one pot of synthesis of amphiphilic diblock copolymers poly[alkylpoly(ethylene glycol) phosphate*-b*-alkylpoly(ethylene glycol) lactate-phosphate]s containing a lactic acid unit in the polymer backbone via polycondensation and oxidation reactions. Micelle formation and drug loading and encapsulation efficiency of copolymers were studied.

## 2. Results and Discussion

### 2.1. Synthesis and Characterization of Poly(ethylene glycol)lactate (PEG-LA)

Poly(ethylene glycol)lactate was prepared via transesterification of ethyl lactate with poly(ethylene glycol 600) (Figure 1). The reaction was carried out at a molar ratio of 1:1 at 120 °C.

The structure and composition of the reaction product were confirmed by ^1^H and ^13^C{H}NMR spectroscopy. In the ^1^H NMR spectrum (Appendix A) after 18 h of heating there is no signal for C*H*_3_CH_2_O protons at 1.32 ppm. This revealed that the reaction product did not contain free ethyl lactate. The two doublets at 1.43 and 1.428 ppm in the ^1^H NMR spectrum can be assigned to C*H*_3_CH-protons. Poly(ethylene glycol) lactate consists of the racemic mixture of levo and dextro forms. The multiple ranging from 4.28 to 4.34 ppm can be assigned to CH_3_C*H*-and CH_2_C*H*_2_OC(O) protons. This signal for CH_3_C*H*-proton has to be a quartet, the one for CH_2_C*H*_2_OC(O) protons is a triplet but as the result of overlapping the signal appears as a multiplet. In the ^13^C{H}NMR spectrum (Appendix A) the signal at 14.21 ppm for the *C*H_3_CH_2_O carbon atom disappears. The signals at δ ppm: 20.34, 61.64, 64.40, 66.72, 68.83, 70.28, 72.48, and 175.51 can be assigned to the carbon atoms of methyl group *C*H_3_CH-, HO*C*H_2_-, CH_3_*C*H-, C(O)O*C*H_2_-;-C(O)OCH_2_*C*H_2_O-,HOCH_2_CH_2_O*C*H_2_-, -(O*C*H_2_*C*H_2_O)-, HOCH_2_*C*H_2_O-, *C*=O, respectively. Data from ^13^C{H} NMR spectroscopy confirm the structure of the reaction product. The yield of poly(ethylene glycol 600)lactate is 70.82%, 41.72 g (based on ^1^H NMR spectrum see (Appendix A); PEG 600-17.76 g (0.03 mol). The proposed method for the synthesis of poly(ethylene glycol)lactate allows varying the molar ratio between ethyl lactate and PEG to synthesize copolymers with different lactic acid content.

### 2.2. Synthesis of Poly[poly(ethylene glycol) H-phosphonate-b-poly(ethylene glycol)lactate H-phosphonate]

It is known that the oxygen atom of the secondary hydroxyl group is a weaker nucleophile compared to the oxygen atom of the primary hydroxyl group. On the other hand, it was known that diphenyl H-phosphonate is more reactive compared to dialkyl esters of H-phosphonic acid in transesterification reactions [1]. In this connection, we decided to use diphenyl H-phosphonate as a starting monomer for the preparation of poly[poly(ethylene glycol) H–phosphonate-*b*-poly(ethylene glycol)lactate H-phosphonate] using poly(ethylene glycol 600) and poly(ethylene glycol)lactate as a dihydroxy compound. To the reaction mixture (see Section 2.3) was added diphenyl H-phosphonate. Transesterification was carried out at a temperature of 145 °C in an inert atmosphere without the use of a catalyst, a vacuum of 0.6 mm Hg, and a reaction time of 10 h. In the ^1^H NMR spectrum of the reaction product after 10 h heating (see Appendix A) there are signals at δ: 1.42 ppm, d, ^3^J(H,H) = 7.07 Hz for C*H*_3_-protons in C(O)CH(C*H*_3_)-OH; 1.60 ppm, d, ^3^J(H,H) = 6.83 Hz, -POCH(C*H*_3_)C(O)-; 1.58 ppm, ^3^J(H,H) = 6.83 Hz, -POCH(C*H*_3_)C(O)-(two diastereoisomers); 3.64, s, -C*H*_2_OC*H*_2_; 4.99–5.06 ppm, m, POC*H*(CH_3_)-protons; 6.83–7.21 for aromatic protons; there are four types of P-H protons at δ = 6.95 ppm, d, ^1^J(P,H) = 717.69 Hz. This doublet can be assigned to a phosphorus atom with the following structure: OCH_2_OP(O)(*H*)OCH_2_ in the repeating units [43]; δ = 7.02 ppm, d, ^1^J(P,H) = 729.40 Hz for OCH_2_OP(O)(H)OCH(CH_3_)-; δ = 7.13 ppm, d, ^1^J(P,H) = 734.8 Hz for OCH_2_OP(O)(*H*)OCH(CH_3_)-; δ = 7.16 ppm, d, ^1^J(P,H) = 724.76 Hz for PhOP(O)(*H*)OCH_2_-; 4.13–4.37 ppm, m, OC*H*_2_C*H*_2_OP(O)(H)OCH_2_CH_2_. The data from ^1^H NMR spectroscopy confirm that the secondary hydroxyl group of poly(ethylene glycol)lactate participated in the transesterification reaction and a diblock copolymer is formed. The ^31^P{H}NMR spectrum (Appendix A). of the reaction product shows five types of phosphorus atom at δ = 9.98 ppm with integral intensity (II) = 1.00; 9.30 ppm, II = 0.34; 8.54 ppm, II = 0.02; 7.90 ppm, II = 0.38; 7.34 ppm, II = 0.05; 5.99 ppm, II = 0.06. The signal at δ = 9.98 ppm in the ^31^P NMR spectrum (Appendix A) is a doublet of quintets with ^1^J(P,H) = 717.78 Hz, and ^3^J(P,H) = 9.7 Hz. This signal can be assigned to the phosphorus atom with the following substituents -CH_2_OP(O)(H)OCH_2_-(type **A**); the signal at 9.30 ppm appears as a doublet of quartets with ^1^J(P,H) = 734.76 Hz, and ^3^J(P,H) = 8.57 and 9.05 Hz. This signal can be assigned to the phosphorus atom with the following substituents -CH_2_OP(O)(H)OCH(CH_3_)C(O)-(type **B**). The second doublet of quartets at 7.90 ppm with ^1^J(P,H) = 729.26 Hz, and ^3^J(P,H) = 9.7 and 10.35 Hz. can be assigned to the phosphorus atom with the same substituents as in the case of the first doublet of quartets. These data revealed that poly(ethylene glycol)lactate exists as a racemic mixture of D- and L-lactides [44]. It is a two-diastereoisomer. The ratio between the integral intensity of the signals at 9.98 ppm, 9.30 ppm, and 7.90 ppm is 1:0.72. The signal at 5.99 ppm which appears as a doublet of triplets with ^1^J(P,H) = 721.41 Hz can be assigned to the phosphorus atom in the end group PhO-P(O)(H)-OCH_2_ (type **C**). The signals at 8.54 ppm with very low intensity 0.02 represents a doublet of triplets and can be assigned to the phosphorus atom with the following substituents -HO-P(O)(H)-OCH_2_-(type **D**), which occurs as a result of partial hydrolysis of the terminal phenoxy group (20% of the terminal PhO group is hydrolyzed). Based on the data from ^1^H and ^31^P NMR spectroscopy, we assume that the reaction of poly(ethylene glycol)lactate and poly(ethylene glycol) with diphenyl H-phosphonate resulted in the formation of diblock copolymer with the following structure (Figure 2).

Based on the ^31^P{H}NMR spectrum (see Appendix A) the value of q = 12, z = 9, the average molecular weight of the reaction product is 14,124 Da, the content of polymer **I** is 80%, that of polymer **II** is 20%. SEC analysis showed that the average number molecular weight of the reaction product after 10 h heating is 13,475 with PDI = 1.64.

The hydrophilicity of the poly[alkylpoly(ethylene glycol) phosphate] **III** is determined by the poly(ethylene glycol) and the phosphoryl group, which is a strong proton acceptor [45]. The hydrophilicity of the poly[alkylpoly(ethylene glycol)lactate phosphate] **IV** will be determined not only by polyethylene glycol and the phosphoryl group but also by the carbonyl group of lactic acid, which is a proton acceptor, too. By varying the ratio between q and z we can control the hydrophilicity of the copolymer and control the hydrophilic/hydrophobic balance of the copolymer (Figure 3).

### 2.3. One-Pot Synthesis of Poly[alkylpoly(ethylene glycol) phosphate-b-alkylpoly(ethylene glycol)lactate phosphate]s

#### One-Pot Synthesis of Poly[hexadecylpoly(ethylene glycol) phosphate)-b-hexadecylpoly(ethylene glycol)lactate phosphate]

Poly[(alkylpoly(ethylene glycol) phosphate-*b*-alkylpoly(ethylene glycol)lactate phosphate] was obtained via one-pot synthesis avoiding a lengthy separation process and purification of the intermediate chemical compound. For this purpose, poly[poly(ethylene glycol H-phosphonate-*b*-poly(ethylene glycol)lactate H-phosphonate] was converted into the corresponding poly[poly(ethylene glycol) chlorophosphate-*b*-poly(ethylene glycol)lactate chlorophosphate] via treatment with trichloroisocyanuric acid (Figure 4).

The conversion was monitored by ^31^P{H}NMR spectroscopy. ^31^P{H}NMR spectrum of the reaction product (Appendix A) after 14 h heating revealed that the signals for the phosphorus atoms of poly[(ethylene glycol H-phosphonate-*b*-poly(ethylene glycol)lactate H-phosphonate] disappear and new three signals at δ = 5.89 ppm; δ = 5.48 ppm and δ = 4.76 ppm appear. These signals can be assigned to the phosphorus atoms of the poly[poly(ethylene glycol) chlorophosphate-*b*-poly(ethylene glycol)lactate chlorophosphate]s. The signal at δ = 5.89 ppm can be assigned to the phosphorus atom in the following structure -CH_2_O*P*(O)(Cl)OCH_2_-. Signals at δ = 5.48 ppm and δ = 4.76 can be assigned to phosphorus atoms with the following structure -CH_2_O-*P*(O)(Cl)OCH(CH_3_)-. The chirality of the CH carbon atom is a reason for appearing of two signals. The ratio between the integral intensity of the signals at 5.89 ppm and (5.49 ppm + 4.76 ppm) is 1: 0.74. This ratio for the integral intensities of the phosphorus atoms of poly[poly(ethylene glycol) H-phosphonate-*b*-poly(ethylene glycol)lactate H-phosphonate] was 1:0.72. This result indicates that the reaction is not accompanied by the occurrence of side reactions. After completion of the oxidation reaction solution of 1-hexadecanol in diethyl ether was added to the reaction product. The formation of poly[hexadecylpoly(ethylene glycol) phosphate-*b*-hexadecylpoly(ethylene glycol)lactate phosphate] was controlled by ^31^P{H} NMR spectroscopy. The reaction is stopped when signals for phosphorus atoms of poly[poly(ethylene glycol) chlorophosphate-*b*-poly(ethylene glycol)lactate chlorophosphate] disappear in the ^31^P{H}NMR spectrum, and new signals at 0.24 ppm, −0.26 ppm and −1.32 ppm (Appendix A) appear, which are characteristic for phosphate structures—-CH_2_O-P(O)(OR)OCH_2_O- and –CH_2_O-P(O)(OR)-OCH(CH_3_)-. Using a one-pot synthesis strategy we synthesized poly[ocylpoly(ethylene glycol) phosphate-*b*-ocylpoly(ethylene glycol)lactate phosphate] and poly[dodecylpoly(ethylene glycol) phosphate-*b*-dodecylpoly(ethylene glycol)lactate phosphate]. The application of a one-pot synthesis strategy for the preparation of poly[alkylpoly(ethylene glycol) phosphate-*b*-alkylpoly(ethylene glycol)lactate phosphate]s has a big advantage because poly[poly(ethylene glycol) chlorophosphate-*b*-poly(ethylene glycol)lactate chlorophosphate]s are highly reactive polymers and extremely sensitive to moisture, so purification requires an absolutely dry atmosphere and dry chemicals. Using one-pot synthesis strategy poly[alkylpoly(ethylene glycol) phosphate-*b*-alkylpoly(ethylene glycol)lactate phosphate]s were obtained in situ, without isolation of the corresponding poly[poly(ethylene glycol) chlorophosphate-*b*-poly(ethylene glycol)lactate chlorophosphate]s.

Table 1 shows the results of molecular weight measurements of the novel polymers using SEC. PDI values ranged from 1.56 to 1.65. The molecular weight of the polymer also increased with increasing side chain length. Therefore, it was suggested that the molecular weight of the polymer can be controlled by the type of side chain used.

### 2.4. Self-Assembly of Poly[alkylpoly(ethylene glycol) phosphate-b-alkylpoly(ethylene glycol)lactate phosphate]s

The results obtained revealed that the newly synthesized polymers self-assembled into the micellar structure in an aqueous solution and micelles show monodisperse peaks (Figure 1). It was found that the particle size is inversely proportional to the increase in the number of hydrophobic side chains (Table 2). This is probably because the increased hydrophobic interaction increased the cohesive force in the inner core of the micelle and reduced the particle size. Therefore, it was suggested that the particle size can be controlled by changing the hydrophobic side chain. In addition, the micelles prepared in this study were dispersed in a size of 10–200 nm, which shows the EPR effect.

### 2.5. Particle Size Distribution and Loading Rate and Encapsulation Efficiency of Sudan III Encapsulating Micelles

Results revealed that the newly synthesized polymers can encapsulate Sudan III (Table 3) and the particle size distribution is monodispersed (Figure 2). It was found that the encapsulation of Sudan III increased in particle size. As in the case of the free micelles, the increase in the number of hydrophobic side chains leads to a reduction in the size of the micelles.

Drug loading and encapsulation efficiency of the newly synthesized polymers relative to Sudan III are low (Table 4). The highest drug loading and encapsulation efficiency revealed poly[hexadecylpoly(ethylene glycol) phosphate-*b*-hexadecylpoly(ethylene glycol)lactate phosphate] 0.75% and 8.16%, respectively. The results obtained revealed that the drug loading and encapsulation efficiency can be improved by introducing a more hydrophobic side chain.

The release of drugs from block copolymer micelles depends upon the rate of diffusion of the drug from the micelles, micelle stability, and the rate of biodegradation of the copolymer. The release rate is mostly influenced by the following factors: the strength of the interactions between the drug and the core-forming block [46], the physical state of the micelle core [47], the amount of drug loaded [46], the molecular volume of the drug, the length of the core-forming block, the molecular weight of copolymer [48] and the localization of the drug within the micelle [49]. Drug release is highly influenced by where the drug molecules are located. If the drug is located predominantly in the corona, then the length of the core-forming block, the micelle size, and the molecular volume of the drug are less important in determining the release rate. On the other hand, the amount of drug loaded in the micelle core; the higher the concentration of drug, the slower the release rate [50,51]. It has been found that the more soluble compounds are localized in the inner corona or the core–corona interface while the more hydrophobic compounds are situated mostly in the micelle core [49]. The release rate of the drug is a function of its localization within the micelle. The outer corona region of the micelle is quite mobile; as a result, release from this area is rapid. The release of drugs localized in the corona or at the interface is said to account for “burst release” from the micelle [49]. The release behavior of Sudan III is shown in Figure 3. The release rate decreases with increasing the alkyl chain length. The micelles release for 12 h 28%, 28%, and 20% Sudan III for poly[octylpoly(ethylene glycol) phosphate-*b*-octylpoly(ethylene glycol)lactate phosphate], poly[dodecylpoly(ethylene glycol) phosphate-*b*-dodecylpoly(ethylene glycol)lactate phosphate)], and poly[hexadecylpoly(ethylene glycol) phosphate-*b*-hexadecylpoly(ethylene glycol)lactate phosphate)], respectively. The release rate of Sudan III strongly decreases after 12 h. For 120 h the released amount of Sudan III was 58.7%, 51.9, and 47.0% for C8, C12, and C16, respectively.

### 2.6. Particle Size Distribution and Drug Loading and Encapsulation Efficiency for Doxorubicin

Micelles of newly synthesized polymers can encapsulate doxorubicin in PBS solution (Table 5). The diameter of the micelles with encapsulated doxorubicin increases compared to the free micelles and those with encapsulated Sudan III. The particle size distribution was monodispersed (Figure 4) and the size was sufficient to exhibit the EPR effect. It can be assumed that selective delivery to tumors is possible as a micelle formulation.

The drug loading efficiency increased with the ratio of drug to polymer and hydrophobic chain lengths of a block copolymer (Table 6). The highest drug loading and encapsulation efficiency were obtained for poly[hexadecylpoly(ethylene glycol) phosphate-*b*-hexadecylpoly(ethylene glycol)lactate phosphate]—1.63% and 45.8%, respectively. Drug encapsulation efficiency is an important index for drug delivery systems, especially for an expensive drug. Drug loading and encapsulation efficiency were higher than those of Sudan III.

Doxorubicin release profiles in PBS at 37 °C (Figure 5) revealed that the release rate of the encapsulated doxorubicin decreases with increasing alkyl chain length and amount of encapsulated doxorubicin. The obtained results are in line with what has been found in the literature that micelles with longer hydrophobic blocks showed slower drug release rates compared with micelles with shorter hydrophobic blocks, as a consequence of increased hydrophobic interaction between the drug molecules and hydrophobic blocks [52].

The doxorubicin release profile revealed that for 12 h the release amount of doxorubicin is: 24.6% for C8; 19.7% for C12 and 18.5% for C16; for 120 h is 55.3% for C8; 53.2% for C12 and 47% for C16. These results obtained showed the influence of the molecular weight of the polymer on the release rate of doxorubicin from micelles, which matches with the established by Kim and coworkers [48] that increase in the molecular weight of copolymer resulted in a decrease in the release rate. The release profile suggests the good stability of DOX-loaded micelles under physiological conditions.

Release profiles of Sudan III and doxorubicin revealed that release proceeds in two stages: first stage—with a higher release rate from 0 h to 12 h; second stage—with a lower rate from 12 h to 120 h. It can be assumed that the decrease in the release rate of Sudan III and doxorubicin is due to their location in the micelle. A higher release rate implies that they are located in the core–corona interface or the periphery of the core, while a lower release rate means they are located in the center of the core. When drug molecules are predominantly located in the core the higher the concentration of the drug, the slower the release rate [48]. The micelles prepared from poly[alkylpoly(ethylene glycol) phosphate]-*b*-poly[alkylpoly(ethylene glycol)lactate phosphate]s entrap doxorubicin and disperse to a size exhibiting the EPR effect. The drug loading was as high as 1.4–1.6% for 3.8% preparation. Based on the above results, the new micelles prepared are considered to be useful as micellar preparations, not only because they are degradable but also because they have a certain degree of sustained release.

## 3. Experimental Part

### 3.1. Materials

Poly(ethylene glycol)s with a number-average molecular weight of 600 g/mol was purchased from Wako Pure Chemical Industries, Ltd. (current FUJIFILM Wako Pure Chemical Co., Osaka, Japan). It was dried before use by a two-stage process: an azeotropic distillation with toluene and a subsequent 4 h heating at 120 °C under a dynamic vacuum. Ethyl lactate was purchased from Wako Pure Chemical Industries, Ltd. It was distilled before use. Sodium methoxide was purchased from Wako Pure Chemical Industries, Ltd. and was used as received. Diphenyl H-phosphonate was purchased from Wako Pure Chemical Industries, Ltd. with, a purity of 85%, containing < 15% phenol. Phenol was removed by distillation before use. 1-octanol, 1-dodecanol, and 1-hexadecanol were purchased from Wako Pure Chemical Industries, Ltd. Trichloroisocianuric acid, 97% (TCIA) Sigma-Aldrich (St. Louis, MO, USA).

### 3.2. Characterization Methods and Instruments

All ^1^H, ^13^C NMR spectra were recorded on a Bruker NMR spectrometer operating at 600 MHz at 37 °C in CDCl_3_. The molecular weights and polydispersity index (PDI) of polymers were determined by size exclusion chromatography (SEC) system (SIL-20A, RID-10A, LC-20AD, CTO-20A, and DGU-20A3; Shimadzu Corp., Kyoto, Japan) with columns (GPC KF-804Lx2; Showa Denko K. K., Tokyo, Japan) using the PEG standard. THF, 10 mM at 40 °C was used as eluent (flow rate: 1.0 mL/min).

### 3.3. Synthesis of Poly(ethylene glycol)lactate

Poly(ethylene glycol) with molecular weight 600 Da (52.57 g, 0.088 mol) and ethyl lactate (10.4 g, 0.088 mol) and NaOCH_3_ (1.9 × 10^−3^ mol%) were stirred under Ar-atmosphere with a magnetic stirrer in a round-bottom two-necked flask, equipped with a condenser, attached to the vacuum line. The reaction took place at temperatures between 100 and 120 °C for 18 h. The byproduct (ethanol) was removed under normal pressure. Reaction progress was controlled by ^1^H NMR spectroscopy (Appendix A). The end of the reaction was subjected to a vacuum to remove unreacted ethyl lactate. The catalyst was removed by extraction. Reaction products are soluble in water, ethanol, dichloromethane, chloroform, and THF. The product is lightly brown. The yield of poly(ethylene glycol 600)lactate was 70.82%, 41.72 g (based on ^1^H NMR spectrum see Appendix A); PEG 600 −17.76 g (0.03 mol).

^1^HNMR (CDCl_3_), δ (ppm) (Appendix A):1.43 (d, H_a_, C*H*_3_CH, ^3^J(H,H) = 7.07 Hz); 1.428 (d,H_a_, C*H*_3_CH, ^3^J(H,H) = 6.83 Hz); 3.47 (d, H_b_, CH_3_CH(O*H*), ^3^J(H,H) = 4.6 Hz); 3.64, H_c_ s, -C*H*_2_OC*H*_2_-; 3.72 (t, H_d_, HOC*H*_2_-, ^3^J(H,H) = 4.6 Hz); 4.28 to 4.34 area for H_f_, CH_3_C*H*(OH)C-, quartet, and H_e_, CH_2_C*H*_2_OC(O)-, triplet, overlapping of the signals;

^13^C{H}NMR (CDCl_3_), δ (ppm) (Appendix A): 20.34, *C*H_3_CH; 61.64, HO*C*H_2_-; 64.40, CH_3_*C*H-; 66.72, -C(O)O*C*H_2_-; 68.83, -C(O)OCH_2_*C*H_2_O-; 70.28, HOCH_2_CH_2_O*C*H_2_-; 70.51, -(O*C*H_2_*C*H_2_O)-; 72.48, HOCH_2_*C*H_2_O-; 175.51, *C*=O.

### 3.4. Synthesis of Poly[poly(ethylene glycol) H-phosphonate-b-poly(ethylene glycol)lactate H-phosphonate]

To 58.92 g reaction product from 2.2 containing: 41.72 g. An amount of 0.042 mol poly(ethyleneglycol600)lactate and 17.76 g 0.022 mol PEG600 were added to 17.61 g, 0.066 mol diphenyl H-phosphonate under Ar-atmosphere in a round-bottom two-necked flask, equipped with a magnetic stirrer, condenser attached to the vacuum line. The reaction was carried out at temperature of 130 °C for 6 h and at 145 °C for 4 h, vacuum 0.6 mm Hg. The progress of the reaction was monitored by the amount of phenol that evolved. When the evolution of phenol stopped, the system was cooled under argon flow. The product was obtained as a waxy solid. Yield, 40.9 g, 92.2%. Molecular weight based on the ^31^P{H}NMR spectrum (see Appendix A) 13,424 Da. ^1^H NMR (CDCl_3_) (Appendix A) δ, ppm, reaction product after 10 h heating: δ = 1.42 ppm, d, ^3^J(H,H) = 7.07 Hz for -C(O)CH(C*H*_3_)-OH protons; 1.60 ppm, d, ^3^J(H,H) = 6.83 Hz, -POCH(C*H*_3_)C(O)-; 1.58 ppm, ^3^J(H,H) = 6.83 Hz, -POCH(C*H*_3_)C(O)-; 3.64, s, -C*H*_2_OC*H*_2_; 4.99- 5.06 ppm, m, POC*H*(CH_3_)-protons; 6.83–7.21 for aromatic protons; δ = 6.95 ppm, d, ^1^J(P,H) = 717.69 Hz. OCH_2_OP(O)(*H*)OCH_2_; δ = 7.02 ppm, d, ^1^J(P,H) = 729.40 Hz for OCH_2_OP(O)(*H*)OCH(CH_3_); δ = 7.13 ppm, d, ^1^J(P,H) = 734.8 Hz for OCH_2_OP(O)(*H*)OCH(CH_3_); δ = 7.16 ppm, d, ^1^J(P,H) = 724.76 Hz for PhOP(O)(*H*)OCH_2_-; 4.13–4.37 ppm, m, OC*H*_2_C*H*_2_OP(O)(H)OCH_2_CH_2_-.

^31^P{H} NMR (Appendix A) (CDCl_3_), δ ppm, reaction product after 6 h heating: 9.98 ppm with integral intensity (II) = 1.00; 9.30 ppm, II = 0.34; 8.54 ppm, II = 0.02; 7.90 ppm, II = 0.38; 7.34 ppm, II = 0.05; 5.99 ppm, II = 0.06 and 5.68 ppm, II = 0.02. The ratio between the integral intensity of the signals at 9.98 ppm and (9.30 ppm and 7.90 ppm) is 1: 0.72 = 1.38.

^31^P NMR (CDCl_3_) (Appendix A), δ ppm: 9.98 ppm, doublet of quintets with ^1^J(P,H) = 717.78 Hz, and ^3^J(P,H) = 9.7 Hz, CH_2_O*P*(O)(H)OCH_2_; 9.30 ppm, dq, with ^1^J(P,H) = 734.76 Hz, and ^3^J(P,H) = 8.57 and 9.05 Hz, CH_2_O*P*(O)(H)OCH(CH_3_)-. Further, 7.90 ppm, dq, with ^1^J(P,H) = 729.26 Hz, and ^3^J(P,H) = 9.7 and 10.35 Hz, CH_2_O*P*(O)(H)OCH(CH_3_)-.

### 3.5. Synthesis of Poly[alkylpoly(ethylene glycol) phosphate-b-alkylpoly(ethylene glycol)-lactate phosphate]s

#### 3.5.1. Synthesis of Poly[hexadecylpoly(ethylene glycol) phosphate-b-[hexadecylpoly(ethylene glycol)lactate phosphate]

Poly[hexadecylpoly(ethylene glycol) phosphate)*-b*-hexadecylpoly(ethylene glycol)lactate phosphate] was obtained from poly[poly(ethylene glycol) H-phosphonate]-*co*-poly(ethylene glycol)lactate H-phosphonate], trichloroisocyanuric acid and 1-hexadecanol (HD) in a one-pot synthesis. The entire synthesis was carried out under an inert atmosphere. A typical synthetic procedure is described below: poly[(ethylene glycol) H-phosphonate]-*co*-[poly(ethylene glycol)lactate H-phosphonate] 1.58 g, 2.4 × 10^−3^ mol; trichloroisocyanuric acid 0.57 g, 2.4 × 10^−3^ mol. To the stirred reaction mixture in acetonitrile (10 mL) at room temperature was added in one portion a solution of trichloroisocyanuric acid in acetonitrile 21 mL. The reaction mixture was kept for 5 h at 40 °C, 4h at 60 °C, and 5 h at 50 °C. The optimum reaction time of 14 h was determined by observing ^31^P{H} NMR spectroscopy.^31^P{H}NMR spectrum showed signals at 5.89 ppm; 5.48 ppm and 4.76 ppm (Appendix A) which are characteristic of chlorophosphate structures. The ratio between the integral intensity of the signals at 5.89 ppm, 5.49 ppm, and 4.76 ppm is 1: 0.74. To the reaction product poly[poly(ethylene glycol) chlorophosphate-*b*-poly(ethylene glycol)lactate chlorophosphate] without isolation, 1-hexadecanol 0.58 g (2.4 × 10^−3^ mol in 7 mL diethyl ether was added. The optimum reaction time was determined by observing ^31^P{H} NMR spectroscopy. Yield 96%, 2.64 g.

^1^H NMR (CD_3_OD), δ (ppm): 0.89, C*H*_3_CH_2_-(t, ^3^J(H,H) = 5.7 Hz); 1.22–1.26, CH_3_(CH_2_)_14_-CH_2_-(m); 1.48–1.53, CH_3_(CH_2_)_8_-(CH_2_)_6_-CH_2_-,m; 3.66−3.58 (m, CH_2_OCH_2_). 4.14−4.10 (m, -CH_2_OP(O)(OCH_2_(CH_2_)_14_-CH_3_)OCH_2_-).

^31^P{H}NMR (CDCl_3_), δ 0.13 ppm; −0.19 ppm, −0.94ppm, −1.2 ppm, and −12.15 ppm.

#### 3.5.2. Synthesis of Poly[octylpoly(ethylene glycol) phosphate-b-octylpoly(ethylene glycol)lactate phosphate]

Using the same procedure was synthesized: poly[octylpoly(ethylene glycol) phosphate-*b*-octylpoly(ethylene glycol)lactate phosphate]. A typical synthetic procedure: poly[(ethylene glycol) H-phosphonate-*b*-poly(ethylene glycol)lactate H-phosphonate] 1.98 g, 3.0 × 10^−3^ mol; trichloroisocyanuric acid 0.70 g, 3.0 × 10^−3^ mol; 1-octanol-0.39 g, 3.0 × 10^−3^ mol. Yield 94%, 2.97 g.

^1^H NMR (CD_3_OD), δ (ppm): 0.89, C*H*_3_CH_2_- (t, ^3^J(H,H) = 5.7 Hz); 1.22–1.26, CH_3_(C*H*_2_)_6_-CH_2_-(m); 1.48–1.53, CH_3_(CH_2_)_4_-(CH_2_)_2_-CH_2_-,m; 3.66−3.58 (m, -CH_2_OCH_2_-); 4.14−4.10 (m, -CH_2_OP(O)(OCH_2_(CH_2_)_6_-CH_3_)OCH_2_-).

^31^P{H}NMR (CDCl_3_), δ 0.11 ppm; −0.20 ppm, −0.93ppm, −1.2 ppm, and −11.15 ppm.

#### 3.5.3. Synthesis of Poly[dodecylpoly(ethylene glycol) phosphate)-b-dodecylpoly(ethylene glycol)lactate phosphate]

Using the same procedure was synthesized poly[dodecyl poly(ethylene glycol) phosphate-*b*-dodecylpoly(ethylene glycol)lactate phosphate].

A typical synthetic procedure: poly[(ethylene glycol) H-phosphonate-*b*-poly(ethylene glycol)lactate H-phosphonate] 2.08 g, 3.1 × 10^−3^ mol; trichloroisocyanuric acid 0.74 g, 3.1 × 10^−3^ mol; 1-dodecanol-0.58 g, 3.1 × 10^−3^ mol. Yield 94%, 3.29 g.

^1^H NMR (CD3OD), δ (ppm): 0.89, CH_3_CH_2_-(t, ^3^J(H,H) = 5.7 Hz); 1.22–1.26, CH_3_(CH_2_)_10-_CH_2_-(m); 1.48–1.53, CH_3_(CH_2_)_8_-(CH_2_)_2_-CH_2_-,m; 3.66−3.58 (m, -CH_2_OCH_2_-). 4.14−4.10 (m, -CH_2_OP(O)(OCH_2_(CH_2_)_10_-CH_3_)OCH_2_-).

^31^P{H}NMR (CDCl_3_), δ 0.11 ppm; −0.18 ppm, −0.93ppm, −1.2 ppm, and −11.15 ppm.

### 3.6. Measurement of Polymeric Micelle Size

Add 50 mg of the synthesized poly[alkylpoly(ethylene glycol) phosphate-*b*-alkylpoly(ethylene glycol)lactate phosphate]s (referred to as sample) and 20 mL of acetone to the flask and dissolve completely. The tin film was formed. Then, the remaining acetone was completely removed by drying under reduced pressure. A PBS solution of pH 7.4 was added thereto. Then, after ultrasonic irradiation, it was filtered through a 0.2 μm filter. Furthermore, the particle size was measured at 37 °C by dynamic light scattering (DLS) system, ELS-Z2 (Otsuka Electronics Co., Ltd., Hirakata, Japan), to evaluate dispersibility.

### 3.7. Solubilizing Test

#### 3.7.1. Preparation of Sudan III Encapsulating Micelles

Acetone 20 mL was added to 50 mg of the sample and 5 mg of Sudan III to dissolve completely. Thereafter, evaporation was performed to form a thin film, and acetone was completely removed by drying under reduced pressure. Then, a PBS solution was added, followed by ultrasonic irradiation, followed by filtration through a 0.2 μm filter to remove liberated Sudan III. An amount of 1 mL of the Sudan III-encapsulating micelle solution was freeze-dried. Then, it was dissolved in 10 mL of acetone and filtered through a 0.45 μm filter to remove precipitated salts. The absorbance of micellar acetone solution was measured using UV–vis, and Sudan III was quantified from a calibration curve prepared separately. The absorbance of the Sudan III-encapsulating micelle solution was measured with UV–Vis at an absorption wavelength of 500 nm, and the value was defined as the absorbance on day 0 (A0). Thereafter, the mixture was shaken in a hot water bath for predetermined hours (3, 6, 12, 24, 48, 72, 96, 120 h), and released Sudan III was removed with a 0.2 μm syringe filter. After that, the absorbance (An) was measured, and the release rate was calculated from the degree of decrease in absorbance by applying the following Formula (1).
(1)$$Releaserateondayn(\%)=\frac{A_{0}-A_{n}}{A_{0}}\times 100$$
where
A_0_: Absorbance of Sudan III encapsulating micelle on day 0A_n_: Absorbance of Sudan III encapsulating micelle on day n.

#### 3.7.2. Preparation of Doxorubicin-Encapsulating Micelles

Two milligrams of doxorubicin hydrochloride were weighed into a flask, and 0.1 mL of triethylamine was added for dehydrogenation. Amounts of 50 mg of sample and 20 mL of chloroform were added and completely dissolved. Evaporation was performed to form a thin film, and chloroform was completely removed by drying under reduced pressure. A PBS solution was added to it. Then, after ultrasonic irradiation, it was filtered through a 0.2 μm filter to remove liberated doxorubicin. One milligram of the doxorubicin-encapsulating micelle solution was freeze-dried. Then, it was dissolved in 10 mL of chloroform and filtered through a 0.45 μm filter to remove precipitated salts. Then, the absorbance of the micelle chloroform solution was measured using UV–Vis, and doxorubicin was quantified from a separately prepared calibration curve. The absorbance of the doxorubicin-encapsulating micelle solution was measured with UV–Vis at an absorption wavelength of 500 nm, and the value was defined as the absorbance on day 0. Thereafter, the mixture was shaken in a hot water bath for predetermined hours (3, 6, 12, 24, 48, 72, 96, 120 h), and freed doxorubicin was removed with a 0.2 μm syringe filter. After that, the absorbance was measured, and the release rate was calculated from the degree of decrease in absorbance by applying Formula (1) in the same manner as for Sudan III-encapsulating micelles.

## 4. Conclusions

Amphiphilic diblock polyphosphoesters containing lactic acid units in the polymer backbone were synthesized by multistep one-pot polycondensation reactions. One-pot synthesis avoids a lengthy separation process and purification of the intermediate chemical compound. The proposed method for the synthesis of poly(ethylene glycol)lactate allows varying the molar ratio between ethyl lactate and PEG to synthesize copolymers with different lactic acid content. The inclusion of lactic acid leads to an increase in the hydrophilicity of the copolymer due to the carbonyl group. The effects of the polymer composition on micelle formation and stability, and micelle size were studied via DLS. The hydrophilic/hydrophobic balance of these polymers can be controlled by changing the chain lengths of hydrophobic alcohols. Loading capacity and encapsulation efficiency depend on the length of alkyl side chains. Drug loading and encapsulation efficiency tests using Sudan III and doxorubicin revealed that hydrophobic can be delivered by copolymers. The highest drug loading and encapsulation efficiency were obtained when hydrophobic alcohol was used hexadecanol—1.63% and 45.8%, respectively. The results obtained indicate that amphiphilic diblock polyphosphoesters have the potential as drug carriers. In this study, we succeeded in synthesizing novel polymers that can form micelles. To evaluate the safety and usefulness of these polymers in vivo, it will be important to conduct cytotoxicity tests, cell uptake tests, bio-security tests, and pharmacokinetic tests in the future.

## Data Availability

The data presented in this study are available in the Appendix A.

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
