# Peer review of "Synthesis and Characterization of Amphiphilic Diblock Polyphosphoesters Containing Lactic Acid Units for Potential Drug Delivery Applications"

_molecules, 2023, doi:10.3390/molecules28135243_

Round 1

Reviewer 1 Report

In this manuscript, the novel polyphosphoesters containing lactic acid units were prepared. Based on the amphiphilic polymers, the novel nanoparticles were fabricated and used as vehicles for drug delivery. It is an interesting work, and the polymers show potential application in drug delivery. However, some other studies on the polymers and nanoparticles should be provided.

1.      The molecular weights of the novel polymers should be studied with GPC;

2.      The morphology of the nanoparticles should be studied with TEM or SEM;

3.      Being used a drug carrier, the cytotoxicity of these nanoparticles need to be investigated;

4.      The cell uptake of these nanoparticles needs to be studied;

5.      The bio-security of these nanoparticles should be provided;

6.      The drug loading and encapsulation efficiency of the drug-loaded nanoparticles should be provided with the style of ‘mean ± SD’;

7.      The stability of these nanoparticles needs to be studied;

8.      The English needs minor revision;

Minor editing of English language required

Author Response

Response to Reviewer 1 Comments

Point 1. The molecular weights of the novel polymers should be studied with GPC.

Response 1: The molecular weight of phosphorus-containing polymers can be measured by 31P{H}NMR spectroscopy. The molecular weight of the starting poly[poly(ethylene glycol) H-phosphonate-b-poly(ethylene glycol)lactate H-phosphonate] was measured by 31P{H}NMR spectroscopy and SEC. In the case of poly[poly(ethylene glycol) chlorophosphate-b-poly(ethylene glycol)lactate chlorophosphate] 31P{H}NMR spectroscopy revealed that the ratio between the integral intensity of the signals at 5.89 ppm and (5.49 ppm + 4.76 ppm) is 1: 0.74. This ratio for the integral intensities of the phosphorus atoms of the starting poly[poly(ethylene glycol) H-phosohonate-b-poly(ethylene glycol)lactate H-phosphonate] was 1:0.72. Knowing the values of q = 12, and z = 9 can be calculated the average molecular weight. The same applies to poly[alkylpoly(ethylene glycol) phosphate-b-alkylpoly(ethylene glycol)lactate phosphate]s.

Point 2. The morphology of the nanoparticles should be studied with TEM or SEM.

Response 2: TEM seems to be more suitable than SEM for the morphological observation of micelles. However, unlike solid nanoparticles, small-sized micelles such as those in this study are difficult to observe by TEM. Morphological observation of micelles is our future research topic.

Point 3. Being used a drug carrier, the cytotoxicity of these nanoparticles need to be investigated.

Response 3: The subject of this paper was to synthesize novel amphiphilic diblock polyphosphoesters. Detailed biological studies are the next research topic and are beyond the scope of this paper, but this comment is important and is mentioned in the 'Conclusion'.

Point 4. The cell uptake of these nanoparticles needs to be studied.

Response 4: The subject of this paper was to synthesize novel amphiphilic diblock polyphosphoesters. Detailed biological studies are the next research topic and are beyond the scope of this paper, but this comment is important and is mentioned in the 'Conclusion'.

Point 5. The bio-security of these nanoparticles should be provided.

Response 5: The subject of this paper was to synthesize novel amphiphilic diblock polyphosphoesters. Detailed biological studies are the next research topic and are beyond the scope of this paper, but this comment is important and is mentioned in the 'Conclusion'.

Point 6. The drug loading and encapsulation efficiency of the drug-loaded nanoparticles should be provided with the style of ‘mean ± SD’.

Response 6: Thank you for your advice. Following your instructions, we have recalculated Tables 3 and 5 and added the standard deviation to all values in the tables.

Point 7. The stability of these nanoparticles needs to be studied.

Response 7: The stability of the encapsulated material was shown as a release profile. Please see Figures 3 and 5.

Point 8. The English needs minor revision.

Response 8: Thank you for pointing this out. We checked and corrected the spelling and grammar of the manuscript.

Reviewer 2 Report

This article “ Amphiphilic Diblock Polyphosphoesters Containing Lactic Acid Units for Drug Delivery” is overall good and covers important area. The idea seems novel and the experiments were carried out nicely. This work investigated the development of a multistep one pot synthesis of amphiphilic diblock copolymers poly[alkylpoly(ethylene glycol) phosphate-b-alkylpoly(ethylene glycol) lactate-phos- phate]s containing a lactic acid unit in the polymer backbone via polycondensation and oxidation reactions. I suggest the acceptance of this manuscript for publication after these minor corrections:

- Please add the main findings (numbers) to the abstract.

- I suggest these two references to be included in this article: 10.3390/molecules28010306 and 10.3390/polym14173697

- Please add authors perspectives and opinions in the conclusion part.

Minor Editing.

Author Response

Response to Reviewer 2 Comments

Point 1. Please add the main findings (numbers) to the abstract.

Response 1: Thank you for your advice. We added information on the size of the micelles and information on changes in the amount of Sudan III and doxorubicin encapsulated in the micelles to the abstract.

Point 2. I suggest these two references to be included in this article: 10.3390/molecules28010306 and 10.3390/polym14173697

Response 2: These two papers have been added to the “Introduction” as references 41 and 42.

Point 3. Please add authors perspectives and opinions in the conclusion part.

Response 3: We added text to the “Conclusions”.

Point 4. Comments on the Quality of English Language: Minor Editing.

Response 4: Thank you for pointing this out. We checked and corrected the spelling and grammar of the manuscript.

Round 2

Reviewer 1 Report

Additional experiments needed, e.g., TEM, GPC, et.al.

Author Response

Thank you for your advice.

We have added detailed information on size exclusion chromatography (SEC) in Section 2.2. In addition, at the end of Section 3.3, we added the results of molecular weight measurements using SEC.

As for TEM and SEM, we do not have the environment to measure them.